# Prevalence and factors associated with anaemia in children aged 6–24 months living a high malaria transmission setting in Burundi

Jean Claude Nkurunziza[1,2]*, Nicolette Nabukeera-Barungi[3], Joan Nakayaga Kalyango[1,4], Aloys Niyongabo[5], Mercy Muwema Mwanja[1], Ezekiel Mupere[3], Joaniter I. Nankabirwa[1,6]

1 Clinical Epidemiology Unit, Makerere University College of Health Sciences, Kampala, Uganda, 2 Department of Community Medicine, Kamenge University Hospital Center (CHUK), University of Burundi, Bujumbura, Burundi, 3 Department of Paediatrics and Child Health, School of Medicine, College of Health Sciences, Makerere University Kampala, Kampala, Uganda, 4 Department of Pharmacy, Makerere University College of Health Sciences, Kampala, Uganda, 5 Department of Laboratory, Kamenge University Hospital Center (CHUK), University of Burundi, Bujumbura, Burundi, 6 Infectious Diseases Research Collaboration, Kampala, Uganda

* jeanclaudenkuru2012@gmail.com

**Data Availability Statement:** Data of the study involve indirect identifiers (names initials, sex, age, location) and may risk the identification of study

## Abstract

### Background

In very young children, anaemia has been linked to increased morbidity, mortality and poor cognitive development. Although Burundi has a high burden of anaemia, which may be worsened by the high burden of malaria, little is known about the extent of the problem in very young children who are most at risk of severe disease. We estimated the prevalence, and assessed the factors associated with anaemia in children aged 6–24 months using baseline data collected as part of an on-going study evaluating the effect of Micronutrient supplementation on anaemia and cognition among children in high malaria transmission settings in Burundi.

### Methods

Between February and March 2020, surveys were conducted in 498 households within the catchment area of Mukenke Health Center. One child aged 6–24 months was selected per household to participate in the survey. Following written informed consent, we administered a questionnaire to the child's primary caregiver to capture information on child's demographics, nutritional status, food intake, health (status, and morbidity and treatment-seeking practices), as well as the household markers of wealth. A physical exam was conducted, and a blood sample was collected to: 1) assess for presence of plasmodium infection using a rapid diagnosis test; 2) estimate the haemoglobin levels using a portable haemocue machine. A stool sample was also collected to examine for the presence of helminth infections.

### Results

The prevalence of anaemia was 74.3% (95% confidence interval [CI] 61.5%-84.0%), with most of the anaemic study participants classified as having moderate anaemia (59.2%). A

participants. Ethical restrictions may be applied on the data supporting this study in the interest of protecting confidential participant information. To access data, interested and qualified researchers may submit requests to the School of Medicine Ethics Committee at the College of Health Sciences, Makerere University (rresearch9@gmail.com), can also submit demands to the current chairperson of the committee, Prof. Ponsianoo Ocama (ponsiano.ocama@gmail.com) or to the Institutional Ethical Committee on Health Research of the Faculty of Medicine, University of Burundi (faculte_medecine_ub@yahoo.fr).

**Funding:** JCN got a two years doctoral fellowship from the African Union and European Union-Intra-ACP Mobility Partnering for Health Professionals Training in African Universities (P4HPT). JIN is supported by the Fogarty International Center (Emerging Global Leader Award grant number K43TW010365). The funders had no role in study design, data collection and analysis, decision to publish, or preparation of the manuscript.

**Competing interests:** The authors have declared that no competing interests exist.

total of 62 (12.5%) participants had positive malaria rapid diagnosis tests. Factors significantly associated with higher odds of developing anaemia included not receiving deworming medication (adjusted Odd ratio [aOR] = 3.54, 95% CI 1.79–6.99, p<0.001), the child's home location (Mukenke II: aOR = 2.22, 95% CI 1.89–2.62, p<0.001; Mukenke: aOR = 2.76, 95% CI 2.46–3.10, p<0.001 and Budahunga: aOR = 3.12, 95% CI 2. 94–3.31, p<0.001) and the child's age group (Children aged 6–11 months: aOR = 2.27, 95% CI 1.32–3.91, p<0.001). Education level was inversely associated with less odds of anaemia: child's primary care giver with a secondary (aOR = 0.67; 95% CI: 0.47–0.95, p = 0,024) and tertiary education level (aOR = 0.48; 95% CI: 0.38–0.61, p<0.001).

## Conclusion

Anaemia is highly prevalent among young children in high malaria transmission setting. Anaemia is more prevalent among children who not dewormed and those with malaria. To prevent the long-term adverse outcomes of the anaemia in children, policy makers should focus on improving uptake of the deworming and malaria prevention programs, promote preventive interventions and improve the education of women especially in families with very young children.

## Introduction

Anaemia is defined as a condition in which the number of red blood cells or the haemoglobin concentration within them is lower than normal, consequently affecting their capacity to carry sufficient oxygen to meet the body's physiologic needs [1–3]. Anaemia has multiple and interconnected causes. immediate causes of anaemia are inadequate diet intake and recent morbidity [4]. Demographic, socioeconomic status, family structure, water/sanitation, growth, maternal health and recent illnesses have been previously reported to be significantly associated with childhood anaemia [5]. There is strong evidence that children living in low income household are at greater risk of anaemia compared to those with higher income. Tesema et al in a study on prevalence and determinants of severity levels of anaemia among children aged 6–59 months in sub-Saharan Africa found that Children from poorest, poorer, middle and richer household wealth were 1.39 times [AOR = 1.39, 95% CI: 1.33, 1.45], 1.32 times [AOR = 1.32, 95% CI: 1.26, 1.37], 1.20 times [AOR = 1.20, 95% CI: 1.15, 1.25], and 1.15 times [AOR = 1.15, 95% CI: 1.11, 1.20] higher odds of higher level of anaemia compared to children from the richest household wealth, respectively [6]. For Tesema et al children whose mother education level had 1.73 times (AOR = 1.73, 95% CI: 1.60, 1.86) at no formal education, 1.39 times (AOR = 1.39, 95% CI: 1.29, 1.50) at primary education, and 1.27 times (AOR = 1.27, 95% CI: 1.18, 1.36) at secondary education level than children whose mother had a higher level of education [6]. Malaria has also been documented as a major cause of anaemia in tropical regions through haemolysis of infected and uninfected erythrocytes and bone marrow dyserythropoiesis which compromises rapid recovery from anaemia [7].

Anaemia is a common childhood disease and public health threat in developing countries [8, 9]. The overall prevalence of anaemia in children under five is estimated at 43% worldwide, with the biggest burden in sub–Saharan Africa [8]. Anaemia in young children has been linked to a number of negative effects including; poor cognitive function, poor motor development, fatigue, and increased morbidity and mortality [10–13]. A number of causes of anaemia in

children have been documented including inadequate dietary intake of iron and fortified foods [14], hookworm infections, and malaria infection [4, 7, 15–17], most of which are preventable. A study in Thailand reported that anaemic status of the hookworm positive group significantly improved by 2 months after deworming of the hookworm-infected children (Hb = 12.1, 95%CI 5.1–19.2) and to became comparable with the helminth-free control group (Hb = 12.0, 95% CI 9.2–15.7) [18].

Central Africa, where Burundi is located, carries the highest burden of anaemia with 71% of the children classified as anaemic [8]. The country is also highly endemic for malaria, and has a high burden of malnutrition [19, 20]. The 2016 Demographic Health Survey (DHS) showed that among children aged 6 to 59 months; 58% were stunted, 45% were anaemic, and 26.8% had malaria parasites [20]. Although previous reports showed a steady reduction in the overall burden of anaemia in the country between 1990 and 2010 [21], recent reports showed an increase in overall burden in 2017 from 45% in 2010 to 61% in 2017. This increase corresponded with a simultaneous increase of malaria prevalence from 17% to 27%. The estimated prevalence of anaemia in Burundi is way beyond the 40% WHO critical level considered as a public health problem [2, 22].Young children have contributed most to the anaemia burden in Burundi. The 2016/2017 Demographic Health Survey estimated the prevalence of anaemia at 84% and 78% in children aged 6–8 and 9–11 months respectively [22]. Young children are also the age-group with the highest risk of getting anaemia as a complication of malaria, increasing their risk of mortality and long-term adverse effects [7, 23]. Despite this high burden, there is not routine monitoring of anaemia in this high-risk group in Burundi.

In this study, we estimated the burden and factors associated with anaemia in children 6–24 months living in Mukenke Health District, northern Burundi, an area with high malaria transmission Our findings will provide information for policymakers to target interventions focusing on anaemia control in this vulnerable population.

## Methods

### Study design and setting

This was a cross-sectional study using baseline data collected as part of an on-going study evaluating the effect of micronutrient supplementation on anaemia and cognition among children in high malaria transmission setting in Burundi. The baseline data collection was conducted between February and March 2020 in Mukenke health district, Kirundo province. The health district is mainly a rural area with four administrative hills (Budahunga, Butegana, Mukenke, and Mukenke II). The district has approximately 35,882 households according to the 2018 annual health report [24], with the main source of income being agriculture and livestock farming. Approximately 27, 674 (17.9%) of the population in the Mukenke health sub-district are children under five years of age, with children between 6–24 months estimated at 9,276 [24]. Mukenke Health Center is the largest public health facility of Mukenke health district with the catchment area composed of 17 villages. The number of children aged 6–24 months within the catchment area of Mukenke Health Center is estimated at 1,116 [24].

Mukenke Health District is among the 23 districts (out of 46) with high malaria burden in Burundi, having an annual incidence estimated at 450 cases per 1000 population. Elsewhere annual malaria incidence was moderate (250–450 cases per 1000) for 12 health districts, low (100–250 cases per 1000) for 9 health districts and very low (less than100 cases per 1000) for 2 health districts [25]. The Health District also has a high burden of anaemia and malnutrition. According to the 2016/2017 Demographic Health Survey, the prevalence of anaemia was 79.2% and that of stunting was 62.9% in children under 5 years of age [22].

## Study population

Using the community health worker registers, households within the catchment area of Mukenke Health Centre with children aged 6–24 months were identified and 501 of these households were selected by simple random sampling using the registered total number of children aged 6–24 months in Mukenke Health center catchment area (n = 1116) and the computed sample size of 492 participants. We calculated the sampling interval by dividing accessible population by the sample size (1116/492 = 2.27 rounded at 2). Using the community health worker registers in each village (the register lists the children and their households), we selected households to participate in the study using an interval of 2 as listed in the register. Meetings were held with the children's primary care-giver to explain the purpose of the study. Following the discussions, written informed consent for their children to participate in the survey was sought from the care-givers. All children aged 6–24 months in the household were screened for eligibility to join the study. Children were enrolled if they were aged 6–24 months, were permanent residents of the selected household, and their parent/primary care giver had provided written informed consent. If a household had more than one child aged 6–24 months, one of the eligible children was randomly selected using inclusion and exclusion criteria for the study such that only one child per household participated in the study. If a selected household did not have a child in the age group on the day of the survey, we then selected the nearest northern household. Children were excluded if they had a known history of sickle cell anaemia.

## Study procedures

For all eligible children, a detailed questionnaire was administered to the primary caretakers by the study team members. The questionnaire was used to capture information on: 1) the participant's demographics, 2) nutrition status and food intake including who prepares and/or feeds the child, breastfeeding status, duration of breastfeeding, age of starting complementary foods, minimum acceptable diet, and consumption of fortified food in the last 2 weeks; 3) the child's health including history of common childhood diseases (fever, diarrhoea, difficult or hard breathing, cough) in the last two weeks, number of times they have been treated for malaria since birth, accessibility to the nearest health facility, time required to reach the nearest health facility, vitamin A supplementation in the last 6 months, routine deworming in the last 6 months and immunisation status, 4) information on the characteristics of the primary care-giver (age, income, education level and employment), 5) detailed information on the proxy indicators of household wealth like the house structure, source of livelihood, and ownership of the house was collected.

Minimum acceptable diet was the consumption of at least four food groups from a list of seven food groups: grains, roots, and tubers; legumes and nuts; flesh foods (meat, fish, poultry, and liver/organ meat); eggs; vitamin A-rich fruits and vegetables; and other fruits and vegetables. Consuming at least four groups meant that the child had a high likelihood of consuming at least one animal source of food and at least one fruit or vegetable in addition to the staple food (grains, roots, or tubers) [26].

A physical examination was conducted on all enrolled children and included: a general examination (nutritional oedema screening, conjunctival and/or palmar pallor), axillary temperature, weight using a SECA® scale (Seca 813 Hamburg, Germany), length or height using a portable Stadiometer Height-Length Measuring Boards (infant/child ShorrBoard® Maryland, USA), and Mild Upper-arm Circumference (MUAC) using a standard MUAC tape (S0145620 MUAC, Child 11.5 Red/PAC-50). A finger-prick (or a heel prick in the case of children age 6–11 months) was done to obtain a blood sample to assess for *Plasmodium* infection

using a rapid diagnostic test (mRDT), and haemoglobin estimation using a HemoCue® porta-ble photometer [2, 27]. We applied altitude adjustment on haemoglobin measured values for Mukenke altitude [28] based on the midpoint of altitude range (1403.5 m for the 1331−1476 range), we used CDC formula published in 1989 MMWR equation [29] for Hb adjustment (g/L) = [(−0.032 × (altitude × 0.0032808) + 0.022 × (altitude × 0.0032808)2) × 10] = -3.20 g/L. The CDC formula provide a good estimation in assessing Haemoglobin up to 2150 m above the sea level [30].Children who were found to have moderate (haemoglobin values 70–99 g/L) or severe anaemia (haemoglobin values <70 g/L) were referred to the nearest health facility for follow-up or care. We provided a universal container, a plastic bag, a clean non-disinfectant impregnated disposable towel and tissue paper to all children primary caregiver in order to collect morning stool samples. All collected stool samples were examined for geohelminths within one to two hours period.

## Laboratory evaluations

Malaria testing was performed in the field by the trained laboratory technicians, using an mRDT (SD Bioline Malaria Ag Pf/Pan rapid test, Standard Diagnostics, Inc., Yongin Si, Gyeonggi-do Republic of Korea). This is a rapid, qualitative test for the detection of HRP-II (Histidine rich protein II) specific to *Plasmodium falciparum* in human blood. Test kits were used before the expiration date, and were transported and stored according to the recom-mended storage conditions (temperature 4–30˚ C, avoid humidity). The kits were kept in their original packaging at room temperature and were prepared using approximately 5 μl of blood and read according to the manufacturer's instructions. Haemoglobin concentration was assessed using a portable haemoglobinometer (HemoCue Ltd., B-Hemoglobin system (HemoCue AB, Angelholm, Sweden) and estimated to an accuracy of 1 g/dL. Stool samples were examined microscopically for the eggs of intestinal nematodes using the Kato-Katz technique [31, 32].

## Data management and analysis

All data were double-entered using Census and Survey Processing System (CSPro) version 7.3 database, United States Census Bureau, September 2019 [33]. Consistency checks were per-formed, and all discrepancies and queries verified against original paper forms. Anaemia was defined using the World Health Organization (WHO) [2] age-specific thresholds cut off as altitude-adjusted haemoglobin lower than 110g/L. Plasmodium infection was defined as hav-ing positive mRDT results, and the proportion of children with malaria infections was calcu-lated as the number of children with a positive results divided by the total number of children enrolled.

The anthropometric index z-scores for Weight-for-age, Weight-for-height and height-for-age Z-scores were generated after entering child sex, age, weight, height in Emergency Nutri-tion Assessment (ENA) for SMART 2011 tool based on WHO 2006 reference data [34, 35]. If they were less than two standard deviations below the reference mean in the WHO chart, chil-dren were classified as stunted (based on height for age Z score), underweight (based on weight for age Z score) or wasted (based on weight for height Z score) [35].

A household wealth index was created using multiple factor analysis from household head earning income, main type of roof and main type of wall of the house, the cooking place for the members of the household, members leaving in the household and the kind of toilet facil-ity. Households were ranked according to their distribution along the index, which was then divided into quartiles.

All statistical analyses were carried out using Stata 14.0 software (STATA Corporation, Col-lege Station, TX). The outcome of interest was the prevalence of anaemia; which was calculated

as the number of children with altitude-adjusted haemoglobin less than 110g/dL divided by the total number of children tested. Ninety-five percent confidence intervals (CI) were estimated for proportions and standard deviations presented for means after cluster adjustment for administrative hills level.

Using sample size of two proportions and fixing alpha at 5% and power at 80% and estimating the proportion of anaemia in children the lower wealth quintile at 69.3% and higher wealth quintile at 59.4% [22], we needed to enrol a minimum of 492 to answer this objective. Using logistic regression, associations between anaemia and potential associated factors were assessed at bivariate analysis. All variables showing an association at 20% significance level at bivariate analysis. Logical model building using both forward and backward elimination was used to generate minimum adequate model and a 5% significance level was considered significant. Children age and mother's age were found as confounders and were included into the multivariable logistic regression model. Although child sex, relationship to primary care giver, stunting status were not showing an association at 20% significance level at bivariate analysis, we also included them into the multivariable logistic regression final model on basis of apriori knowledge [36–38]. Akaike's Information Criteria (AIC) and Bayesian Information Criteria (BIC) were used to test the model and smaller values of AIC and BIC indicating better model fitting.

### Ethical considerations

We obtained ethical approval from the School of Medicine Higher Degrees Research and Ethics Committee of Makerere University College of Health Sciences (REF #: 2019–079) and the Institutional Ethical Committee on Health Research of the Faculty of Medicine, University of Burundi (Ref #: FM/CE/02/02/2020). The study complied with the national guidelines and got clearances from the Ministry of Health and the Ministry of Intern affaires to be carried out as a household survey. Informed consent was signed by the primary caregivers of the children prior to their interviews and blood and stool sample collection from the children. Participant's data was linked to a code number to ensure confidentiality. After communicating to the mothers/primary caretakers, children diagnosed with moderate/severe anaemia, clinical malaria or intestinal parasite infestation were referred to the health facility for etiological diagnosis and/or management.

## Results

### Characteristics of the study population

A total of 501 were screened for eligibility to join the study of which 498 (99.4%) were enrolled (Fig 1). The reason for exclusion of the 3 children was refusal of the primary care giver to provide written informed consent for the child to participate in the study. The study enrolled slightly more girls (51.2%) than boys, and the mean age at enrolment was 15.24 months (standard deviation [SD] ± 5.74). Majority of the children enrolled had an up-to date status of immunisation (82.1%). A total of 62 (12.5%) participants had positive mRDTs. None of the 329 children tested positive for geohelminth infections on stool examination. Almost all the study children's primary care-giver was their mother (95.5%), with most care-givers having little or no education (66.1%). The details of the characteristics of the study population are presented in Table 1.

### Prevalence of anaemia

The mean haemoglobin concentration was 99.65 g/l (Standard Deviation [SD] 0.69). Of the enrolled children, 370 had haemoglobin less that 110g/L giving an overall estimated prevalence

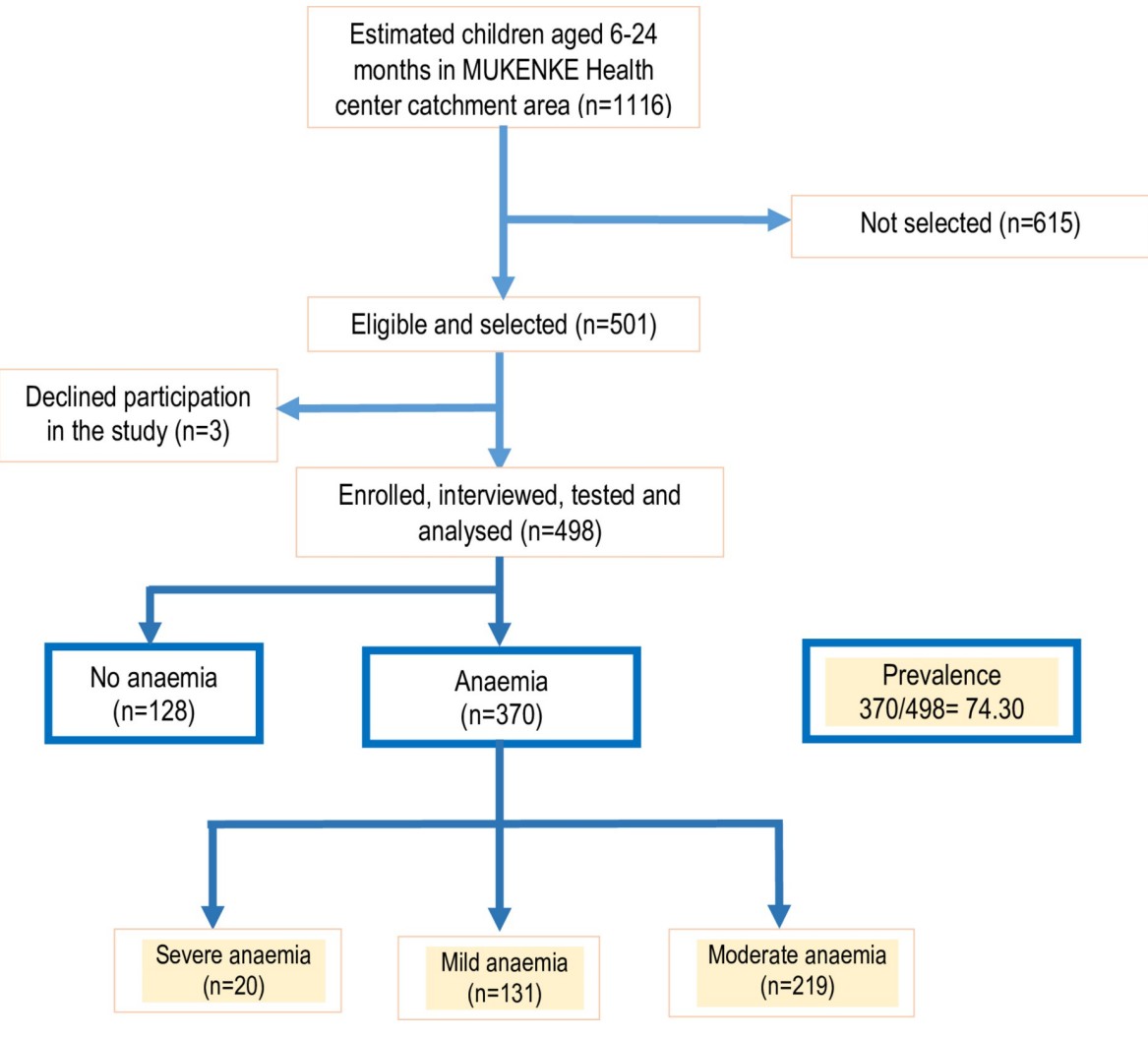

**Fig 1. Participant flow chart.**

of anaemia of 74.3% (95% CI 61.5–84.0). Of the anaemic study participants, 131 (35.4%) had mild anaemia, 219 (59.2%) had moderate anaemia, and 20 (5.4%) had severe anaemia.

Anaemia prevalence was higher in children aged 6-11months compared to the older children (83.4% versus 69.6%, aOR = 2.27, 95% CI:1.32–3.91; p-value = 0.003). The prevalence of anaemia was also higher in children who had a positive mRDT test than those that were negative (87.1% versus 72.5%, p-value = 0.017) (Table 2).

## Factors associated with anaemia in the study population

At multivariate analysis, factors significantly associated with anaemia included: 1) the administrative location of the child's home; 2) status of deworming; 3) child's age group; and 4) Education level of the child primary care giver. Our study showed that children from Mukenke II, Mukenke and Budahunga had 2 to 3 times higher odds of having anaemia than children from Butegana (aOR = 2.19; 95% CI: 1.82–2.63; aOR = 2.82; 95% CI: 2.51–3.16 and aOR = 3.20; 95% CI: 3.07–3.34 respectively). Children aged 6 to 11 months 2 times higher odds of developing anaemia compared to children aged 12 to 24 months (aOR = 2.27; 95% CI:1.32–3.91). Children

**Table 1. Characteristics of the study population.**

| Characteristic | Number (n) | Percentage (%) |
|---|---|---|
| Sex | | |
| Male | 243 | 48.8 |
| Female | 255 | 51.2 |
| Age category | | |
| 6–11 m | 169 | 33.9 |
| 12–24 m | 329 | 66.1 |
| Administrative hill (location) | | |
| Budahunga | 120 | 24.1 |
| Butegana | 64 | 12.9 |
| Mukenke | 164 | 32.9 |
| Mukenke II | 150 | 30.1 |
| Immunisation status Up-to date | | |
| No | 89 | 17.9 |
| Yes | 409 | 82.1 |
| Breastfeeding status | | |
| Never breast-fed | 13 | 2.6 |
| Stopped breastfeeding | 81 | 16.3 |
| Still breastfeeding | 404 | 81.1 |
| Mothers age | | |
| <20 | 69 | 14.7 |
| 20–40 | 382 | 81.3 |
| >40 | 19 | 4.0 |
| Relationship with primary care giver | | |
| Mother | 476 | 95.6 |
| Other relative | 22 | 4.4 |
| Education level of primary care giver | | |
| None/Primary | 329 | 66.1 |
| Secondary | 147 | 29.5 |
| Tertiary | 22 | 4.4 |
| Number of siblings under 5 years of age | | |
| 0–1 | 321 | 64.5 |
| More than 1 | 177 | 35.5 |
| Social economic status of the household | | |
| Very poor | 124 | 25.5 |
| Poor | 119 | 24.5 |
| Middle | 121 | 25.0 |
| Rich | 121 | 25.0 |
| Diet includes fortified foods | | |
| No | 465 | 93.4 |
| Yes | 33 | 6.6 |
| Taken deworming medication in last 6 months | | |
| No | 44 | 8.8 |
| Yes | 285 | 57.2 |
| Not applicable | 169 | 33.9 |
| Wasted | | |
| No | 415 | 83.5 |
| Yes | 82 | 16.5 |

(*Continued*)

**Table 1.** (Continued)

| Characteristic | Number (n) | Percentage (%) |
|---|---|---|
| Stunted | | |
| No | 264 | 53.0 |
| Yes | 234 | 47.0 |
| Underweighted | | |
| No | 337 | 67.8 |
| Yes | 160 | 32.2 |
| Malaria test results (RDT) | | |
| Negative | 436 | 87.6 |
| Positive | 62 | 12.5 |

who didn't get deworming medication in the last 6 months had higher odds of having anaemia compared to those who took deworming medication (aOR = 3.54; 95% CI: 1.79–6.99). Children whose primary care giver have a secondary and tertiary education level had less odds of having anaemia (aOR = 0.68; 95% CI: 0.45–0.99 and 0.43; 95% CI: 0.33–0.58) than others.

## Discussion

This study estimated the burden and factors associated with anaemia among children aged 6 to 24 months in Mukenke Health District, northern Burundi, an area with high malaria transmission. The overall prevalence of anaemia was 74.3% with the burden highest in children under one year of age. Factors associated with anaemia included the child's deworming status, the child's age group, location of the child's home, and education level of the child primary care giver. These study findings highlight the urgent need to address the anaemia burden in this setting in order to reduce on the future possible effects in the population.

The prevalence of anaemia in the study is alarmingly high, with almost three quarters of the study population meeting the definition of being anaemic. Most worrying is that the children under one year of age were more likely to be anaemic than the older children, and majority of the anaemic children had moderate to severe anaemia. The observed burden in this setting is of concern given that anaemia in this age-group may lead to delayed growth and impaired neurological function [39, 40]. Although the causes of anaemia were not evaluated as part of the study, given the study setting, a number of factors may be attributed to the high burden observed. The possible causes of anaemia in the children in this setting may include nutritional origins especially iron deficiency which is the most common cause of childhood anaemia in African children [27].

Other causes of anaemia in this population could be infectious diseases such as malaria and HIV among others [36]. Indeed, malaria may be a key contributor to the observed burden given that the study was conducted in a high malaria burden setting. Malaria infection causes haemolysis of infected and uninfected erythrocytes and bone marrow dyserythropoiesis resulting in reduced haemoglobin concentration [7, 41]. Unfortunately, the greatest impact is in young children, and in particular infants [42]. Even if the p-value was not significant, our bivariate analysis showed that children with a positive malaria test have more than two times the odds of a having anaemia than children with a negative test. A number of other studies have demonstrated the relationship between malaria and anaemia for example, a study by Legason et al in North-western Uganda showed that anaemia was 1.5 times higher among malaria positive children compared to malaria negative [43]. On the other hand, helminth infections which have been documented to cause anaemia are also common parasitic

**Table 2. Factors associated with anaemia among children aged 6–24 months.**

| Characteristic | n/N (% of anaemia) | Crude OR (95% CI) | p-value | Adjusted OR (95% CI) | p-value |
|---|---|---|---|---|---|
| Sex | | | | | |
| Male | 183/243 (75.72) | 1 | | 1 | |
| Female | 186/255 (72.94) | 0.86 (0.58–1.29) | 0.478 | 0.72 (0.43–1.20) | 0.209 |
| Age category (months) | | | | | |
| 12–24 | 229/329 (69.60) | 1 | | 1 | |
| 6–11 | 141/169 (83.43) | 2.20 (1.31–3.68) | 0.001 | 2.27 (1.32–3.91) | 0.003 |
| Administrative hill (location) | | | | | |
| Butegana | 38/64 (59.38) | 1 | | 1 | |
| Budahunga | 101/120 (84.17) | 3.64 (1.81–7.32) | <0.001 | 3.20 (3.07–3. 34) | <0.001 |
| Mukenke | 120/164 (73.17) | 1.87 (1.02–3.42) | 0.004 | 2.81 (2.51–3.16) | <0.001 |
| Mukenke II | 111/150 (74.0) | 1.95 (1.05–3.61) | 0.035 | 1.19 (1.82–2.63) | <0.001 |
| Mother's age | | | | | |
| 20–40 | 278/382 (72.77) | 1 | | 1 | |
| <20 | 55/69 (79.71) | 1.47 (0.74–2.94) | 0.276 | 1.23 (0.72–2.11) | 0.439 |
| >40 | 16/19 (84.21) | 1.99 (1.11–3.60) | 0.021 | 1. 75 (0.74–4.13) | 0.095 |
| Relationship to primary care giver | | | | | |
| Mother | 355/476 (74.58) | 1 | | 1 | |
| Other relative | 22/498 (68.18) | 0.73 (0.32–1.69) | 0.463 | 2.71 (0.33–22.42) | 0.356 |
| Education level of primary care giver | | | | | |
| None/Primary | 254/329 (77.20) | 1 | | 1 | |
| Secondary | 103/147 (70.7) | 0.69 (0.49–0.97) | 0.034 | 0. 67 (0.47–0.95) | 0.024 |
| Tertiary | 13/22 (59.09) | 0.43 (0.29–0.64) | 0.000 | 0.48 (0.38–0.61) | <0.001 |
| Number of siblings <5 years of age | | | | | |
| 0–1 | 235/321 (73.21) | 1 | | 1 | |
| More than 1 | 135/177 (76.27) | 1.20 (0.96–1.45) | 0.125 | 1.17 (0.91–1.51) | 0.211 |
| Social economic status | | | | | |
| Very poor | 91/124 (73.39) | 1 | | 1 | |
| Poor | 88/119 (73.95) | 1.02 (0.58–1.82) | 0.921 | 0.95 (0.41–1.23) | 0.910 |
| Middle income | 94/121 (77.69) | 1.26 (0.70–2.27) | 0.434 | 1.17 (0.48–2.84) | 0.729 |
| Rich | 86/121 (71.07) | 0.89 (0.51–1.56) | 0.686 | 0.94 (0.54–1.62) | 0. 823 |
| Diet includes fortified foods | | | | | |
| Yes | 28/33 (84.85) | 1 | | 1 | |
| No | 342/465 (73.55) | 0.50 (0.14–1.82) | 0.290 | 0. 42 (0.15–1.21) | 0.108 |
| Dewormed in last 6 months | | | | | |
| Yes | 190/285 (66.7) | 1 | | 1 | |
| No | 39/44 (88.6) | 3.90 (1.49–10.22) | 0.006 | 3.54 (1.79–6.99) | <0.001 |
| Not applicable | 141/169 (83.4) | 2.52 (1.57–4.05) | <0.001 | - | - |
| Stunted | | | | | |
| No | 193/264 (73.11) | 1 | | 1 | |
| Yes | 177/234 (77.64) | 0.66 (0.51–0.86) | 0.002 | 0.90 (0.60–1.37) | 0.582 |
| Under weighted | | | | | |
| No | 242/337 (71.81) | - | | 1 | |
| Yes | 128/160 (80.0) | 0.64 (0.47–086) | 0.003 | 0.77 (0.61–1.07) | 0.131 |
| Malaria test results | | | | | |
| Negative | 316/436 (72.48) | 1 | | 1 | |
| Positive | 54/62 (87.10) | 2.56 (0.18–5.55) | 0.017 | 2.49 (0.89–6.96) | 0.136 |

infections in the tropics and subtropics [36]. Helminths interrupt the host's acquisition of nutrients by ingestion and digestion of host blood resulting mainly into iron deficiency [44]. We found that deworming was associated with reduced odds of having anaemia, which is in agreement with the mechanism of action, given the treatment eliminates the helminth infections. Indeed, studies have widely reported the relationship between deworming and anaemia [45]. Although out of the scope of this work, inherited forms of anaemia especially sickle cell anaemia have been documented to cause anaemia, the commonest in this setting being sickle cell aneamia [46].

High prevalence of anaemia in the very young children has been documented in similar settings in Africa. In Ethiopia, the prevalence of anaemia in children 6–23 months was estimated at 71% in 2005, 61% in 2011, and 72% in 2016 [47]. In Namutumba district in Uganda, the prevalence was estimated at 58.8% [48]. In Armenia the highest prevalence of anaemia (67·9%) was detected among 6–12-month-old children [49]. According to the 2016/2017 DHS in Burundi, the prevalence of anaemia in same age-group was estimated at 79.2% showing only a slight reduction in estimates over the last 4 years [22].

The iron stores present at birth and in breast milk protect the infant from iron deficiency up to 6 months of age. But due to the rapid growth of children in the first months of life, there is increased micronutrient requirements including iron. At this age, although breast milk provides some nutrients, it is not sufficient to sustain his growth and development [50]. Above this age, it is well known that dietary sources of iron become critical to keep up with the child's rapid rate of red blood cell synthesis [50], and infants should start receiving additional sources of iron to maintain sufficient haemoglobin concentration. Ensuring that their nutritional needs are met therefore requires to start complementary feeding at 6 months and complementary foods must be adequate, safe and properly fed. Food fortification and universal or targeted nutrient supplementation may also help to ensure that older infants and young children receive adequate amounts of iron and other micronutrients [51].

In addition to the child's age group and taking deworming medication, location of the child's home where the child resides and education level of primary care givers were associated with anaemia. Our study showed a statistical association between anaemia and administrative location of the child's home. Children from Mukenke and Budahunga were almost 3 times more likely to be anaemic than children from Butegana. This could be explained by the fact that Budahunga hosts a site of a population of the Batwa ethnic (population group which has marginalized nutrition habits) and for Mukenke, the presence of single-parent families abandoned by fathers who were seasonal workers in the area artisanal mining. Khan et al. in Bangladesh revealed similar association; childhood anaemia was significantly associated with geographical location defined by division 60.4% among children under the age of 5 were anaemic in Barisal and 58.9% in Rangpur division while the lowest was 47.8% in Dhaka division [52]. This is also similar to what was observed by Sharma et al. in India; a significant spatial heterogeneity in the prevalence of anaemia among children [53]. By contrast, our findings are not consistent with other analysis where they didn't find this association [5, 54].

Anaemia was negatively associated with education level of the child primary care giver. Children whose mothers had a secondary and tertiary education level were less likely to be anaemic. For Tesema et al children whose mother education level had 1.73 times at no formal education, 1.39 times at primary education, and 1.27 times at secondary education level than children whose mother had a higher level of education [6]. The level of education of primary care givers may positively influence health care and feeding practices of their children. Educated mothers are more aware of their children's health and take into account the nutritional values of foods. Educated mother understand better and provide a healthy and hygienic diet for their children, improving their nutritional status [55, 56].

Our findings also showed an association between deworming and anaemia. Watthanakul-panich et al, in their study in Thailand reported that anaemic status of the hookworm positive group significantly improved by 2 months after deworming of the hookworm-infected children (Hb = 12.1) and to became comparable with the helminth-free control group (Hb = 12.0) [18]. On the other hand, a study on effect of deworming on school-aged children's physical fitness, cognition and clinical parameters in a malaria-helminth co-endemic area of Ivory Cost revealed that deworming showed no effect on haemoglobin levels and anaemia. The mean Hb levels did not differ significantly between surveys (baseline, 120.4 g/l; follow-up, 119.9 g/l; p = 0.631) and the overall prevalence of anaemia remained unchanged (baseline, 34.3%; follow-up, 34.3%) [57]. Despite the variations in the above findings, it is known that repeated chemotherapy at regular intervals (periodic deworming) in high-risk groups can ensure that the levels of infection are kept below those associated with morbidity [58] and intervention studies have shown positive associations between mass deworming and decreased prevalence of anaemia among children from developing countries [54]. The deworming with two rounds of albendazole and praziquantel spaced by 2 months is highly efficacious against soil-transmitted helminths among children [59].

## Study limitations

Our study has some limitations that should be taken into consideration. A cross sectional design of the study limits our ability to assess temporal or causal of the associated variables, and the study did not aim to identify the aetiology of anaemia. Our study did not measure other known risk factors like maternal anaemia, HIV disease, haemoglobinopathies and what are the levels and how they would affect our results.

## Conclusions

Our study showed a very high burden of anaemia among children aged 6–24 months in Mukenke Health district with the prevalence much higher than the WHO cut-offs for declaring the burden a major public health problem. We found that having a primary care giver with secondary and tertiary education level were associated with reduced odds of anaemia while not taking deworming medication, very young children and a residence location were associated with higher odds. To prevent the long-term adverse outcomes of anaemia, policy makers should focus on improving uptake of deworming, malaria prevention programs, as well as promote women education and preventive interventions of families with very young children.

## Supporting information

**S1 File. Multivariable logistic regression model.**
(PDF)

**S2 File. Study questionnaire English.**
(PDF)

**S3 File. Study questionnaire local language (Kirundi).**
(PDF)

## Acknowledgments

The authors are thankful to the MoH and Ministry of interior, community development and public security for granting us the official permission to conduct the study at community level.

The authors are grateful to communal administrator of Bwambarangwe and all staff of Mukenke Health District, Kirundo for their collaboration in study implementation.

## Author Contributions

**Conceptualization:** Jean Claude Nkurunziza, Nicolette Nabukeera-Barungi, Aloys Niyongabo, Joaniter I. Nankabirwa.

**Data curation:** Jean Claude Nkurunziza.

**Formal analysis:** Jean Claude Nkurunziza, Joan Nakayaga Kalyango, Mercy Muwema Mwanja, Joaniter I. Nankabirwa.

**Investigation:** Jean Claude Nkurunziza.

**Methodology:** Jean Claude Nkurunziza, Nicolette Nabukeera-Barungi, Joan Nakayaga Kalyango, Mercy Muwema Mwanja, Joaniter I. Nankabirwa.

**Software:** Jean Claude Nkurunziza, Joaniter I. Nankabirwa.

**Supervision:** Jean Claude Nkurunziza, Nicolette Nabukeera-Barungi, Joan Nakayaga Kalyango, Aloys Niyongabo, Ezekiel Mupere, Joaniter I. Nankabirwa.

**Validation:** Jean Claude Nkurunziza, Nicolette Nabukeera-Barungi, Joan Nakayaga Kalyango, Aloys Niyongabo, Mercy Muwema Mwanja, Ezekiel Mupere, Joaniter I. Nankabirwa.

**Visualization:** Jean Claude Nkurunziza, Joan Nakayaga Kalyango, Joaniter I. Nankabirwa.

**Writing – original draft:** Jean Claude Nkurunziza, Nicolette Nabukeera-Barungi, Joan Nakayaga Kalyango, Aloys Niyongabo, Mercy Muwema Mwanja, Joaniter I. Nankabirwa.

**Writing – review & editing:** Jean Claude Nkurunziza, Nicolette Nabukeera-Barungi, Joan Nakayaga Kalyango, Aloys Niyongabo, Mercy Muwema Mwanja, Ezekiel Mupere, Joaniter I. Nankabirwa.

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
