## [Decision Letter · Decision Letter 0]

5 Jan 2022

PONE-D-21-22573

Prevalence and factors associated with anaemia in children aged 6-24 months living a high malaria transmission setting in Burundi.

PLOS ONE

Dear Dr. %Jean Claude Nkurunziza%,

Thank you for submitting your manuscript to PLOS ONE. After careful consideration, we feel that it has merit but does not fully meet PLOS ONE’s publication criteria as it currently stands. Therefore, we invite you to submit a revised version of the manuscript that addresses the points raised during the review process.

Please respond to all the comments made

We look forward to receiving your revised manuscript.

Kind regards,

Mary Hamer Hodges, MBBS MRCP DSc

Academic Editor

PLOS ONE

Journal Requirements:

4. Your abstract cannot contain citations. Please only include citations in the body text of the manuscript, and ensure that they remain in ascending numerical order on first mention.

5. Please include a copy of Tables 1 and 2 which you refer to in your text on pages 11 and 12.

Reviewers' comments:

Reviewer's Responses to Questions

**Comments to the Author**

1. Is the manuscript technically sound, and do the data support the conclusions?

Reviewer #1: Yes

2. Has the statistical analysis been performed appropriately and rigorously? 

Reviewer #1: Yes

3. Have the authors made all data underlying the findings in their manuscript fully available?

Reviewer #1: Yes

4. Is the manuscript presented in an intelligible fashion and written in standard English?

Reviewer #1: Yes

5. Review Comments to the Author

Reviewer #1: Reviewer #1: The authors present results of a cross-sectional study that determined the prevalence of and factors associated with anaemia in children 6-24 months living in a high malaria transmission setting in Burundi. They found a high prevalence of anaemia particularly moderate anaemia. Children not dewormed and those with malaria were more likely to be anaemic. The manuscript is well-written and technically sound.

Major comments:

1. Provide more details of systematic sampling procedure referred to in Study Population section (Line 86)

2. Line 168 Data Management and analysis. A little more detail about the cluster adjustment for administrative hills level.

3. Line 177. Sentence appears to be incomplete

4. Line 183 Describe what criterion was used for eliminating models. For instance, AIC, BIC etc,

Minor comments:

5. Abstract/Conclusion. Line 25 and 26. Please revise sentence ‘

6. Introduction: Line 48. Change county to country

7. Line 123 Study Procedures. Change plasmodium to Plasmodium

8. Line 158. Change underweighted to underweight

9. Results Line 221 Change table 2 to Table 2

10. Line 228- 229 MUKENKE, BUDAHUNGA and BUTEGANA should all be written in Sentence case

11. Line 304, 306 and 341. Same comments as above

12. Conclusions Line 348. Delete in

6. PLOS authors have the option to publish the peer review history of their article (what does this mean?). If published, this will include your full peer review and any attached files.

Reviewer #1: No

---

## [Author Response · Author response to Decision Letter 0]

12 Feb 2022

February 10, 2022

To: Academic Editor

PLOS ONE

Reference number: PONE-D-21-22573

Thank you very much for the careful consideration of our manuscript and for inviting us to re-submit our revised manuscript (based on the Reviewers' comments) entitled: “Prevalence and factors associated with anaemia in children aged 6-24 months living a high malaria transmission setting in Burundi ” Below are point by point responses to the reviewers’ comments. We are re-submitting a revised version of the manuscript that addresses the points raised during the review process. We look forward to your favourable consideration of this re-submission. 

POINTS BY POINTS RESPONSES TO THE REVIEWERS’ COMMENTS

The revised manuscript was adapted to PLOS ONE’s style requirements

Study questionnaire in English is available on protocols.io: dx.doi.org/10.17504/protocols.io.b4ryqv7w

Study questionnaire in local language (Kirundi) is available as attached file under materials section on protocols.io: dx.doi.org/10.17504/protocols.io.b4vxqw7n

Data of the study involve indirect identifiers (names initials, sex, age, location) and may risk the identification of study participants. Ethical restrictions may be applied on the data supporting this study in the interest of protecting confidential participant information. To access data, interested and qualified researchers may submit requests to the School of Medicine Ethics Committee at the College of Health Sciences, Makerere University (rresearch9@gmail.com), can also submit demands to the current chairperson of the committee, Prof. Ponsianoo Ocama (ponsiano.ocama@gmail.com) or to the Institutional Ethical Committee on Health Research of the Faculty of Medicine, University of Burundi (faculte_medecine_ub@yahoo.fr).

4. Your abstract cannot contain citations. Please only include citations in the body text of the manuscript, and ensure that they remain in ascending numerical order on first mention.

Ok, no citations in abstract.

5. Please include a copy of Tables 1 and 2 which you refer to in your text on pages 11 and 12.

Table 1 on pages 12-13 and Table 2 on pages 13

Reference list corrected and completed.

Major comments:

1. Provide more details of systematic sampling procedure referred to in Study Population section (Line 86)

Response:

Using the computed sample size of 492 participants, and the registered total number of children aged 6-24 months in Mukenke Health center catchment area (n=1116), we calculated the sampling interval by dividing accessible population by the sample size (1116/492=2,27 rounded at 2). Using the community health workers (CHW) registers in each village (the register lists the children and their households), we selected households to participate in the study using an interval of 2 as listed in the register. We selected only one participant (child aged 6-24 months) per household using inclusion and exclusion criteria (page 6 lines 89-99).

2. Line 168 Data Management and analysis. A little more detail about the cluster adjustment for administrative hills level.

Response:

We anticipate that clustering exists at the administrative hills level and we adjusted for this by including administrative hills as a covariate to the final model. The final adjusted model of significant covariates was 

Logit (anaemia) = 1.89 + 1.08 Budahunga +1.01 Mukenke+ 1.40 No deworming +0.81 Tertiary Ed -0.81 poor SES -0.96 Middle SES- 0.86 Rich SES

[Stata command: logit anaemia i.chsex ib2.ChageGpe ib2.rezhill ib1.BreastFdSt ib2.MothAge ib1.Deworming i.CgChrelat i.Under5sibl i.CgEducGp i.SES3quintile3 ib1.csfpfood Stunting Wasting2 Underweight i.malarscrn, vce(cluster rezhill) ]

3. Line 177. Sentence appears to be incomplete

All variables showing an association at 20% significance level at bivariate analysis and potential confounders were considered for inclusion into the multivariable logistic regression model (page 10 lines 183-184).

4. Line 183 Describe what criterion was used for eliminating models. For instance, AIC, BIC etc,

Response:

For eliminating models, we used Akaike’s Information Criteria (AIC) and Bayesian Information Criteria (BIC). Smaller values of AIC and BIC indicating better model fitting (page 10 lines 192-194).

Minor comments:

5. Abstract/Conclusion. Line 25 and 26. Please revise sentence ‘

Sentence revised (page 3 line 27)

6. Introduction: Line 48. Change county to country

county changed to country (page 4 line 49)

7. Line 123 Study Procedures. Change plasmodium to Plasmodium

plasmodium changed to Plasmodium (page 8 line 129)

8. Line 158. Change underweighted to underweight

underweighted changed to underweight (page 9 line 164)

9. Results Line 221 Change table 2 to Table 2

table 2 changed to Table 2 (page 13 line 230)

10. Line 228- 229 MUKENKE, BUDAHUNGA and BUTEGANA should all be written in Sentence case

MUKENKE, BUDAHUNGA and BUTEGANA written in Sentence case: Mukenke, Budahunga and Butegana (page 15 lines 237- 238)

11. Line 304, 306 and 341. Same comments as above

Sentence case of Mukenke and Budahunga (page 18 lines 312), Butegana (page 18 line 314), Mukenke (page 20 line 349)

12. Conclusions Line 348. Delete in

in deleted (page 20 line 357)

---

## [Decision Letter · Decision Letter 1]

22 Mar 2022

PONE-D-21-22573R1Prevalence and factors associated with anaemia in children aged 6-24 months living a high malaria transmission setting in BurundiPLOS ONE

Dear Dr. %Nkurunziza%,

Thank you for submitting your manuscript to PLOS ONE. After careful consideration, we feel that it has merit but does not fully meet PLOS ONE’s publication criteria as it currently stands. Therefore, we invite you to submit a revised version of the manuscript that addresses the points raised during the review process.

Please address each comment from the Reviewer

We look forward to receiving your revised manuscript.

Kind regards,

Mary Hamer Hodges, MBBS MRCP DSc

Academic Editor

PLOS ONE

Reviewers' comments:

Reviewer's Responses to Questions

**Comments to the Author**

1. If the authors have adequately addressed your comments raised in a previous round of review and you feel that this manuscript is now acceptable for publication, you may indicate that here to bypass the “Comments to the Author” section, enter your conflict of interest statement in the “Confidential to Editor” section, and submit your "Accept" recommendation.

Reviewer #2: (No Response)

2. Is the manuscript technically sound, and do the data support the conclusions?

Reviewer #2: Yes

3. Has the statistical analysis been performed appropriately and rigorously? 

Reviewer #2: No

4. Have the authors made all data underlying the findings in their manuscript fully available?

Reviewer #2: Yes

5. Is the manuscript presented in an intelligible fashion and written in standard English?

Reviewer #2: Yes

6. Review Comments to the Author

Reviewer #2: I have placed main concerns as introductory statement in the review which I have uploaded. The concerns are mainly on altitude adjustment, taking clustering of the data into account during analysis and the association between deworming and anaemia being chance effect as it can be verified from absence of any infection among the study children.

7. PLOS authors have the option to publish the peer review history of their article (what does this mean?). If published, this will include your full peer review and any attached files.

Reviewer #2: No

---

## [Author Response · Author response to Decision Letter 1]

5 May 2022

We are re-submitting a revised version of the manuscript that addresses the points raised during the second review. The specific file description is 'Response to Reviewers'.

---

## [Decision Letter · Decision Letter 2]

15 Jun 2022

PONE-D-21-22573R2Prevalence and factors associated with anaemia in children aged 6-24 months living a high malaria transmission setting in BurundiPLOS ONE

Dear Dr. %Nkurunziza%,

Thank you for submitting your manuscript to PLOS ONE. After careful consideration, we feel that it has merit but does not fully meet PLOS ONE’s publication criteria as it currently stands. Therefore, we invite you to submit a revised version of the manuscript that addresses the points raised during the review process. Please address each point remaining made by reviewer number 2.

We look forward to receiving your revised manuscript.

Kind regards,

Mary Hamer Hodges, MBBS MRCP DSc

Academic Editor

PLOS ONE

Journal Requirements:

Reviewers' comments:

Reviewer's Responses to Questions

**Comments to the Author**

1. If the authors have adequately addressed your comments raised in a previous round of review and you feel that this manuscript is now acceptable for publication, you may indicate that here to bypass the “Comments to the Author” section, enter your conflict of interest statement in the “Confidential to Editor” section, and submit your "Accept" recommendation.

Reviewer #1: All comments have been addressed

Reviewer #2: (No Response)

2. Is the manuscript technically sound, and do the data support the conclusions?

Reviewer #1: Yes

Reviewer #2: Partly

3. Has the statistical analysis been performed appropriately and rigorously? 

Reviewer #1: Yes

Reviewer #2: Yes

4. Have the authors made all data underlying the findings in their manuscript fully available?

Reviewer #1: Yes

Reviewer #2: Yes

5. Is the manuscript presented in an intelligible fashion and written in standard English?

Reviewer #1: Yes

Reviewer #2: Yes

6. Review Comments to the Author

Reviewer #1: (No Response)

Reviewer #2: Dear Prof. Mary Hamer Hodges,

Thank you for inviting me to review the manuscript entitled “Prevalence and factors associated with anaemia in children aged 6-24 months living a high malaria transmission setting in Burundi”. The authors have addressed most of the comments given in the 1st round of my review except very few. If the authors address the following comments, the manuscript can be accepted. The comment numbers are the comments indicated in my previous review comment.

Methods

Comment 4: what are the specific variables included to generate the wealth index (Line 161-

62)? Are the variable all numeric (i.e., quantitative)? If all variables are quantitative, go with

the PCA, qualitative only use the MCA and mixed type, it is advisable to use multiple factor

analysis.

This part of my comment is not well addressed. PCA is used for dimension reduction of continuous variables, multiple correspondence analysis (MCA) for categorical data and if the variables used to reduce the dimension of the data is mixed (i.e both categorical and continuous) like the case in the current manuscript, multiple factor analysis is used. Thus, I recommend to use MFA instead of PCA in the current manuscript.

Comment 5: In line 165 through 167, and also in the abstract section, hemoglobin level is

adjusted for altitude but the data is collected by questionnaire and double entered into a

software. How you get the altitude of the household for the children involved in the study.

The response provided by the authors for this comment is not satisfactory. I recommend adjusting hemoglobin for altitude based on reference Haemoglobin concentrations for the diagnosis of anaemia and assessment of severity published by WHO

Results

Comment 4: The absence of any parasitic infections from the kato-katz technique indicates that

young children were not at risk for such infections. The association between deworming

and anaemia seems chance effect than a true effect due to not being dewormed.

Is there any causal pathway that deworming can increase the hemoglobin concentration in the absence of any parasitic infections? I think it is not the deworming drug that influenced the hemoglobin concertation, rather it may be the supplementations given to the children during the deworming campaign. If this was not the case, I think it is better to remove deworming from the analysis.

Discussion

Comment 1: I think the discussion should not be crowded by numbers for comparison. Such

numbers on the prevalence of different study findings shall be presented in the introduction

section. And make the comparisons based on the level of significance and looking your

confidence intervals than putting the numbers. The same thing is true for effect size of the

factors. This comment is misunderstood by the authors. I recommend not to use the numbers which are indicated in the introduction and result section of the manuscript. Remove the odds ration and its 95% CI in the discussion section.

7. PLOS authors have the option to publish the peer review history of their article (what does this mean?). If published, this will include your full peer review and any attached files.

Reviewer #1: No

Reviewer #2: No

---

## [Author Response · Author response to Decision Letter 2]

30 Jul 2022

Response to review comments for the manuscript entitled: “Prevalence and factors associated with anaemia in children aged 6-24 months living a high malaria transmission setting in Burundi” Ref: PONE-D-21-22573R2.

Thank you very much for the careful consideration of our manuscript and for inviting us to re-submit our revised manuscript (based on the second round of Reviewers' comments) entitled: “Prevalence and factors associated with anaemia in children aged 6-24 months living a high malaria transmission setting in Burundi” Below are point by point responses to the reviewers’ comments. We are re-submitting a revised version of the manuscript that addresses the points raised during the second review. We look forward to your favourable consideration of this re-submission. 

POINTS BY POINTS RESPONSES TO THE REVIEWERS’ COMMENTS

Methods 

Comment 1: what are the specific variables included to generate the wealth index (Line 161-

62)? Are the variable all numeric (i.e., quantitative)? If all variable are quantitative, go with

the PCA, qualitative only use the MCA and mixed type, it is advisable to use multiple factor

analysis. Our response: “Variables used in the SES included: household head earning income, house ownership, house wall, main source of energy for cooking, main type of floor and main source of drinking water. This detail is included in the manuscript (line 186-189). ”: Comment to response: This part of my comment is not well addressed. PCA is used for dimension reduction of continuous variables, multiple correspondence analysis (MCA) for categorical data and if the variables used to reduce dimension of the data is mixed (i.e both categorical and continuous) like the case in the current manuscript, multiple factor analysis is used. Thus, I recommend to use MFA instead of PCA in the current manuscript.

Response: Thank you for this in-put, we have now used the MFA instead of PCA to generate the wealth index. From the new analysis, socioeconomic status was no longer significant in the final model and has been dropped. The updated results are presented in Table 1 (Page 13-14) and Table 2 (Page 15) as well as the text (line 200-1203). 

Comment 2: In line 165 through 167, and also in the abstract section, hemoglobin level is adjusted for altitude but the data is collected by questionnaire and double entered into a

software. How you get the altitude of the household for the children involved in the study. Our response: We applied altitude adjustment on haemoglobin measured values for Mukenke altitude based on the midpoint of altitude range (1403.5 m for the 1331−1476 range), we used 1989 MMWR equation for Hb adjustment (g/L) = [(−0.032 × (altitude × 0.0032808) + 0.022 × (altitude × 0.0032808)2) × 10] = -3.20 g/L (line 147-150). Comment to response: The response provided by the authors for this comment is not satisfactory. I recommend adjusting hemoglobin for altitude based on reference. Haemoglobin concentrations for the diagnosis of anaemia and assessment of severity published by WHO available at the following URL.

https://apps.who.int/iris/bitstream/handle/10665/85839/WHO_NMH_NHD_MNM_11.1_eng.pdf?sequence=22&isAllowed=y

Response: WHO-based cutoffs for Children 6 - 59 months of age are: Non- anaemia: 110 g/l or higher, 100-109 g/l for mild, 70-99 g/l for moderate and lower than 70 g/l for severe anaemia. While altitude adjustments to measured haemoglobin concentrations is -2g/l for 1000m metres above sea level and -5g/l for 1500 metres above sea level. In this study we didn’t not collect the altitude estimates for the homes of each child, however, we used the midpoint altitude estimates of the study area (Mukenke altitude ranges between 1331−1476 and we used 1403.5 m for the average estimate)as reported in elevation map . We used the CDC formula published in 1989 MMWR “equation for Hb adjustment (g/L) = [(−0.032 × (altitude × 0.0032808) + 0.022 × (altitude × 0.0032808)2) × 10]” = -3.20 g/L , . This detail is included in the manuscript, line 160-164.

Comment 3: Absence of any parasitic infections from the kato-katz technique indicates that

the young children were not at risk for such infections. The association between deworming

and anaemia seems chance effect than true effect due to not dewormed. Is there any causal pathway that deworming can increase the hemoglobin concentration in the absence of any parasitic infections? I think it is not the deworming drug that influenced the hemoglobin concertation, rather it may be the supplementations given to the children during the deworming campaign. If this was not the case, I think it is better to remove deworming from the analysis.

Response: The association between deworming and anaemia in very young children has previously been well documented in clinical trials . Our study showed that for the 329 stool samples tested, none was positive for geohelminth infections. Of these, 285 (86.6%) had received deworming medication during the December 2019 national deworming campaign and 169 (33.9%) were aged 6 to 11 months and at low risk for geohelminth infections. From this analysis we are confident that absence of any parasitic infections from the kato-katz technique in those at risk of infection could be also a result of the deworming campaign. The only additional supplementation received during the campaign was vitamin A which may not explain the association observed in the study.We have recoded the covariate “Deworming” and the analysis now takes into account the very young children (deworming not applicable). We still find a statistically significant relation in the bivariate and the multivariate analysis between anaemia and deworming: adjusted Odd ratio [aOR] of 3.54 and 95% CI 1.79-6.99, p<0.001 for 44 children who didn’t not receive deworming medication last six months out of 329 who were at risk of geohelminth infections (Table 2).

Comment 4: I think the discussion should not be crowded by numbers for comparison. Such

numbers on the prevalence of different study findings shall be presented in the introduction

section. And make the comparisons based on the level of significance and looking your

confidence intervals than putting the numbers. The same thing is true for effect size of the

factors. This comment is misunderstood by the authors. I recommend not to use the numbers which are indicated in the introduction and result section of the manuscript. Remove the odds ratio and its 95% CI in the discussion section.

Response: Thank you for this suggestion. We proceeded as recommended: We presented numbers on the prevalence of different study findings shall be presented in the introduction section (line 45-55, 65-68) and we removed the odds ratios and 95% CI in the discussion section (lines 305, 342-343 and 355). 

References:

 Elevation map. Elevation map for localities. Elevation of Mukenke, Bwambarangwe, Burundi (Latitude: 2.58 South, Longitude: 29.95 East, Altitude: 1331.00m/4599.74ft and Latitude: 2.58 South, Longitude: 30.32 East, Altitude: 1476.00m/4842.52ft). Elevation map for localities 2021.

 Centers for Diseases Control and Prevention. CDC criteria for anemia in children and childbearing-aged women. MMWR Morbidity and mortality weekly report. 1989; 38(22):400-4.

 Sarma H, Wangdi K, Tariqujjaman M, et al. The Effects of Deworming and Multiple Micronutrients on Anaemia in Preschool Children in Bangladesh: Analysis of Five Cross-Sectional Surveys. Nutrients 2021; 14(1).

 Bauleni A, Tiruneh FN, Mwenyenkulu TE, et al. Effects of deworming medication on anaemia among children aged 6-59 months in sub-Saharan Africa. Parasit Vectors 2022; 15(1): 7.

---

## [Decision Letter · Decision Letter 3]

15 Aug 2022

Prevalence and factors associated with anaemia in children aged 6-24 months living a high malaria transmission setting in Burundi

PONE-D-21-22573R3

Dear Dr. %Nkurunziza%,

We’re pleased to inform you that your manuscript has been judged scientifically suitable for publication and will be formally accepted for publication once it meets all outstanding technical requirements.

Kind regards,

Mary Hamer Hodges, MBBS MRCP DSc

Academic Editor

PLOS ONE

Additional Editor Comments (optional):

Reviewers' comments:

Reviewer's Responses to Questions

**Comments to the Author**

1. If the authors have adequately addressed your comments raised in a previous round of review and you feel that this manuscript is now acceptable for publication, you may indicate that here to bypass the “Comments to the Author” section, enter your conflict of interest statement in the “Confidential to Editor” section, and submit your "Accept" recommendation.

Reviewer #2: All comments have been addressed

2. Is the manuscript technically sound, and do the data support the conclusions?

Reviewer #2: Yes

3. Has the statistical analysis been performed appropriately and rigorously? 

Reviewer #2: Yes

4. Have the authors made all data underlying the findings in their manuscript fully available?

Reviewer #2: Yes

5. Is the manuscript presented in an intelligible fashion and written in standard English?

Reviewer #2: Yes

6. Review Comments to the Author

Reviewer #2: All my concerns are corrected. However, a language edition may be necessary to improve the overall quality of the paper.

7. PLOS authors have the option to publish the peer review history of their article (what does this mean?). If published, this will include your full peer review and any attached files.

Reviewer #2: No

---

## [Editor Report · Acceptance letter]

24 Aug 2022

PONE-D-21-22573R3 

Prevalence and factors associated with anaemia in children aged 6-24 months living a high malaria transmission setting in Burundi. 

Dear Dr. Nkurunziza:

I'm pleased to inform you that your manuscript has been deemed suitable for publication in PLOS ONE. Congratulations! Your manuscript is now with our production department. 

Kind regards, 

on behalf of

Prof. Mary Hamer Hodges 

Academic Editor

PLOS ONE